# Analysis of Declarative and Procedural Knowledge According to Teaching Method and Experience in School Basketball

**María G. Gamero** [1,2], **Juan M. García-Ceberino** [1], **Sergio J. Ibáñez** [1,3] and **Sebastián Feu** [1,2,*]

1   Optimisation of Training and Sports Performance Research Group (GOERD), University of Extremadura, 10003 Cáceres, Spain; mgamerob@alumnos.unex.es (M.G.G.); jmanuel.jmgc@gmail.com (J.M.G.-C.); sibanez@unex.es (S.J.I.)
2   Faculty of Education, University of Extremadura, 06006 Badajoz, Spain
3   Faculty of Sports Science, University of Extremadura, 10003 Cáceres, Spain
*   Correspondence: sfeu@unex.es

**Abstract:** Analysing declarative and procedural knowledge in sport makes it possible to evaluate the students' acquisitions in the learning process. This study aimed to compare the acquisition of declarative and procedural knowledge after the implementation of several intervention programmes in school basketball, according to the methodology and prior experience of the students. A total of 55 students from the sixth year of primary education took part in the study, distributed into three groups. Each group participated in a different intervention programme: tactical games approach (TGA), direct instruction (DI) or service teacher's basketball unit (STBU). The level of knowledge was measured using the Test of Declarative and Procedural Knowledge in Basketball (TDPKB). A descriptive analysis was performed to determine the participants' characteristics. A factorial ANOVA was subsequently applied in two phases (pre-test and post-test) for independent samples to compare the level of knowledge among the different groups, and a t-test for related samples was performed to compare the pre–post knowledge level within each group. Then, a factorial ANOVA and a test of repeated measures were carried out to determine the effect of the methodology and experience on the students' knowledge. The results indicate that the TGA, DI and STBU intervention programmes induced improvements in the levels of declarative and procedural knowledge in all the groups, with the students who participated in the TGA programme achieving higher levels of declarative knowledge. Finally, the effect of the absence of practical experience was identified as a determining factor for improvement. The students who had not previously practised basketball achieved higher levels of knowledge with the TGA intervention programme.

**Keywords:** basketball; experience; learning; physical education; sport; teaching methodologies





## 1. Introduction

Sports initiation represents the first contact with sports practice and is the period in which the player gets to know and becomes familiar with the sport. For this to occur, there must be situations in which the player learns, develops and perfects the abilities and skills corresponding to each sport. In the context of physical education, invasion sports are the ones most commonly used by teachers who choose the sports disciplines that help them to best attain the set educational objectives [1].

Within this extensive group of sports, basketball represents an invasion sport in which there is the simultaneous participation of several players in space and time through the relation of cooperation and opposition [2]. Its use in the school context is justified because it foments personal relations, perceptive and decisional processes, the acquisition of values and motor development. Moreover, open skills predominate in its practical implementation, thus increasing the implication of the student's perceptual and decisional mechanisms [3].

The teachers play a fundamental role in sports planning as they are able to design effective tasks that permit the achievement of the set objectives [4]. Task planning is of

great importance as, based on acquired knowledge and prior experience, they embody all the intentions for the development of the objectives and implicitly and explicitly implement the methodological concepts held by the coach/teacher on sports teaching [5]. It should thus be the result of a reflective process and not the product of the coach/teacher's improvisation [6]. It is the first phase of action in the quality cycle of their intervention. Moreover, in academic research on the coach/teacher, one of the most commonly chosen topics is the study of their practical behaviour and the teaching style [7].

Regarding methodological concepts, the traditional methodology must be highlighted, which has been questioned in recent years for the teaching of team sports, and non-linear pedagogy. This emerges as a more flexible alternative, focusing its attention on the learner and their needs. It understands the teaching process as a dynamic system in which subject and environment are in continuous interaction [8]. Therefore, there are two main approaches taken by the physical education teacher for teaching invasion sports [9]: the teacher centred approach, (hereinafter TCA) based on traditional methodology, and the student centred approach, (hereinafter SCA) based on nonlinear pedagogy. Within the TCA, the direct instruction (DI) methodology is the most common, while in SCA methodologies, the tactical games approach (TGA) stands out, among others [10].

In the case of the TCA methodology, the teacher proposes the tasks to develop movement patterns that the student has to acquire through repetition, generating little cognitive and motor implication on the part of the student [9]. It is based on the acquisition of individual technical abilities using practical analytical situations, where non-specific tasks are prioritised, isolated from actual play, to be later incorporated into the game [4,11]. Under this approach, the teacher uses prescriptive feedback to correct mistakes [9].

However, the foundation of SCA methodologies is teaching based on the game and they promote the use of teaching styles that involve the student cognitively, making use of guided discovery and problem solving, where the teacher leads the teaching–learning process [9]. The teacher presents a tactical problem that has to be developed through a series of tasks or games, and designs meaningful and contextualised situations in the actual game that foment the students' learning [9,12]. The type of feedback used is interrogative, seeking the students' reflection to achieve more meaningful learning [13]. In short, it pursues greater involvement and participation on the part of the students so that they understand the nature of the game and improve their decision-making tactics [14].

The teacher's intervention in the teaching process can be analysed using observational systems to collect information such as the Integral Analysis System of Training Tasks (SIATE) [5]. This tool makes it possible to analyse the organisational characteristics of the tasks designed by the teachers and coaches, as well as the pedagogical and external load variables that define them, providing information on the teacher's methodological approach [15].

Current advances in the assessment of invasion sports help physical education teachers to draw solid conclusions on their interventions during the teaching process [16]. Scientific evidence is increasingly conclusive on the advisability of using SCA methods rather than a TCA methodology in sports teaching [17,18], as improvements are identified in cognitive-type variables related with declarative knowledge [19], decision making, tactics awareness and the learning of technical components [20]. The evaluation of the acquisitions achieved by the subjects after the teaching–learning process is carried out using different variables such as knowledge of the sport, effectiveness in technical ability and effectiveness in game performance [21].

The implementation of different teaching–learning methodologies in basketball implies a differentiated response in the load experienced by the students or athletes [22,23]. Moreover, the amount of learning acquired by the students, in procedural knowledge, is also affected [24,25]. Due to its characteristics, the SCA methodology induces more pronounced heart rate levels, regardless of the type of sports that it is practised. This improves the cardiorespiratory fitness of the students [26]. These innovative methodologies allow the levels of physical fitness to improve while developing the motor skills of the student,

being more effective and obtaining higher levels of heart rate in invasion games [27,28]. Therefore, its use is recommended for the design of basketball tasks in physical education, in order to provide students with high intensity sessions, where at least 50% of the time is spent in values of moderate–vigorous physical activity (MVPA) [28]. For this, a reduced game format (small games) is recommended, using small spaces with 2vs2 and 3vs3 game situations.

For sports practice to be effective, the students' level of knowledge and their capacity for making decisions are fundamental [29]. The scientific literature differentiates two types of knowledge: declarative and procedural. According to Anderson [30], declarative knowledge is the concept that someone has of something and is identified with "knowing", "knowing what to say" and "knowing what". This type of knowledge refers to the knowledge athletes have of the skills and strategies of the game, and refers to "knowing what to do" [31]. However, tactical procedural knowledge refers to the performance and creation of movements, to know what to do and when, selecting the most suitable actions depending on the different situations in the game [31,32] referring to "how to do it" [33].

The importance of declarative and procedural knowledge in the acquisition of learning has already been emphasised in previous investigations. The majority of studies have focused on the sports context, focusing its attention on game performance by analysing performance indicators and the effectiveness of technical skills execution and decision-making [14,34], few contributions from the sphere of sports initiation in school [25]. The studies carried out by García-Ceberino et al. [35] on soccer and García and Ruiz [36] on handball focus their attention on the cognitive process of sport through the analysis of declarative and procedural knowledge. However, the results obtained are not conclusive. In the study on school football, significant improvements were obtained by means of the DI method in both types of knowledge. Meanwhile in the study on extracurricular handball, there are improvements when the teaching model guides towards tactics.

Few studies have analysed declarative and procedural learning in basketball in the school context; therefore, further research is necessary. Most of the studies use professors–researchers for the application of the programs. This study analyses the figure of the physical education teacher, fundamental in the teaching–learning process. Iglesias et al. [37] study the influence of a reflective supervision programme on decision making and performing the pass in a basketball team of 12–13-year-olds, showing significant improvements in the experimental group. González-Espinosa et al. [25] compare the sports learning about the actions of the game, execution and final efficiency acquired by two groups of students who had received a different intervention programme, identifying significant improvements in favour of the SCA compared to the TCA.

Williams and Davids [38] indicate that declarative and procedural knowledge is acquired through sports practice, differentiating the expert students from the novices in their structured memory. This is why different studies conclude that the level of knowledge and experience of the students are related, with the more experienced students presenting higher levels of knowledge [39,40]. García-Ceberino et al. [35] and Serra-Olivares et al. [41] state that more experienced students recorded better results. Thus, practising a sport outside the school context influences the levels of a student's declarative and procedural knowledge.

Specifically designed instruments are necessary for evaluating declarative and procedural knowledge. Otero et al. [33] designed a tool with 20 items for assessing declarative and procedural knowledge and decision making in school soccer. García-Ceberino et al. [42] proposed the evaluation of learning using the Instrument for Measuring Learning and Performance in Football (IMLPFoot). Several instruments have been used to evaluate students' learning in basketball. The Basketball Learning and Performance Assessment Instrument (BALPAI) [16] is the most complete tool, compared with the previous proposals. It includes 66 items which evaluate a total of 11 play actions, 7 offensive play actions with and without the ball (dribbling; shooting; passing; receiving; passing game; occupying free spaces without the ball; offensive rebound), and 4 on ball and off ball in defence (on

ball defence; off ball defence; defensive help/defensive change; defensive rebound). In addition, Gamero et al. [43] designed the Test of Declarative and Procedural Knowledge in Basketball (TDPKB), which is specifically aimed at assessing the students' level of acquired learning in basketball in the educational context.

There is a scarcity of studies analysing students' acquisition of knowledge in the school context after the implementation of an intervention programme. For this, the aim of this study was to assess the acquisition of declarative and procedural knowledge in primary education students depending on the teaching–learning method used and their prior experience, after the application of different intervention programmes for basketball in the educational context.

## 2. Materials and Methods

### 2.1. Study Design

The present study had a manipulated quasi-experimental longitudinal approach, with a pre-test post-test design to determine the differences in the level of declarative and procedural knowledge after the implementation of three intervention programmes in school basketball [44].

### 2.2. Sample

A total of 55 students of 11 and 12 years from the sixth year of primary education participated in the study. The intervention was carried out in a state school in the west central region of Spain. The students were divided into three heterogeneous mixed groups (6A, 6B and 6C). The administration of the teaching–learning programmes to the groups was random. The students from 6A participated in the direct instruction (DI) intervention, those from 6B were given the tactical game approach (TGA) and 6C participated in the service teacher´s basketball unit (STBU), programme designed and implemented by the school's service physical education teacher, without an explicit definition of the teaching method.

The students had had no contact with the invasion sport of basketball in their physical education classes in previous years. However, Table 1 shows that a high percentage of students did practise basketball as a training/physical activity out of school for two hours a week.

**Table 1.** Characteristics of the students per group and previous experience.

| Methodology and Group | Experience | | | |
| --- | --- | --- | --- | --- |
| | Yes | | No | |
| | *n* | % | *n* | % |
| DI (6A) | 8 | 44.40 | 10 | 55.60 |
| TGA (6B) | 10 | 52.60 | 9 | 47.40 |
| STBU (6C) | 5 | 27.80 | 13 | 72.20 |

Note: *n*, Sample; DI, direct instruction; TGA, tactical games approach; STBU, service teacher´s basketball unit.

### 2.3. Instruments and Material

Learning assessments should be conducted with a validated instrument to ensure that the results obtained are valid and reliable. The declarative and procedural knowledge of the students was analysed using the Test of Declarative and Procedural Knowledge in Basketball (TDPKB) [43]. This instrument is made up of 34 items and was validated by a panel of 15 experts. The TDPKB is valid and reliable for the evaluation of declarative and procedural knowledge of basketball in the school context because it surpassed the critical value ($V > 0.74$) and obtained an excellent internal reliability score ($\alpha = 0.95$).

The first part of the test evaluates the declarative knowledge of the students, understood as the theoretical information about a sport and refers to "what to do". It is evaluated using multiple choice questions.

The second part evaluates procedural knowledge based on the tactical resolution of play situations and refers to "how to do it". It is evaluated using images which represent different game situations, and in this case, there is only one correct answer.

Lastly, the data collected with the TDPKB were exported to the SPSS 24 statistical programme (IBM Corp. Lanzado 2016. IBM SPSS Statistics for Windows, Version 24, IBM Corp, Armonk, NY, USA).

### 2.4. Variables

Two independent variables were determined: (1) the teaching–learning methods and (2) the practice of basketball out of school (prior experience).

Different intervention programmes were taken into account for the methodological variable: the alternative teaching programme for basketball (PEAB) was based on the TGA and the traditional basketball teaching programme (PETB) was based on the DI method [45]. The intervention programmes were equivalent for each methodology in the number of tasks, contents and game phase ($p > 0.05$). The PEAB and PETB programmes were validated by a panel of 17 experts, who found that they were valid and reliable for the teaching of basketball in the school context by surpassing the critical value ($V > 0.70$) and obtaining an excellent internal validity score of ($\alpha = 0.96$) [46]. An external teacher, who was also a researcher in the field of sport pedagogy and basketball coach, implemented these programmes.

The STBU programme was designed and administered by a working physical education teacher with 28 years' experience as a primary teacher. This teacher had total freedom in the design of his didactic unit on the sport of basketball.

The dependent variables of the study were the levels of declarative and procedural knowledge acquired by the students after the application of the intervention programmes.

Table 2 shows the descriptive analysis of each of the intervention programmes according to the following pedagogical variables that define a task: game situation, teaching means, level of opposition and feedback. The definition and categorisation of the variables corresponding to the tasks were performed using the SIATE [5], with an adaptation to the initial proposal [45] in the variables game situation and teaching means to reduce the number of categories. The tasks were categorised by an external evaluator with specific training in the study topic and experience in the use of this tool.

**Table 2.** Descriptive analysis of the didactic units according to the game situation, teaching means, opposition level and feedback.

| Pedagogical Variables | Categories | Methodology/Group | | | | | |
| | | DI (6A) | | TGA (6B) | | STBU (6C) | |
| | | *n* | % | *n* | % | *n* | % |
|---|---|---|---|---|---|---|---|
| Game situation | Without opposition | 29 | 61.70 | - | - | 57 | 86.40 |
| | Individual game | 5 | 10.60 | 16 | 45.70 | 1 | 1.50 |
| | Inequality SSG | 7 | 14.90 | 12 | 34.30 | 7 | 10.60 |
| | Equality SSG | 3 | 6.40 | 2 | 5.70 | 1 | 1.50 |
| | Full game | 3 | 6.40 | 5 | 14.30 | - | - |
| Teaching means | Application exercises | 44 | 93.60 | - | - | 57 | 86.40 |
| | Specific games | - | - | 29 | 82.90 | 5 | 7.60 |
| | Non-specific games | 3 | 6.40 | 4 | 11.40 | 4 | 6.10 |
| | Sport/mini basketball | - | - | 2 | 5.70 | - | - |
| Opposition level | Without opposition | 29 | 61.70 | - | - | 52 | 78.80 |
| | Static obstacle | - | - | - | - | 1 | 1.50 |
| | Dynamic obstacle | 15 | 31.90 | - | - | 4 | 6.10 |
| | Modulated opposition | - | - | 4 | 11.40 | - | - |
| | With opposition | 3 | 6.40 | 31 | 88.60 | 9 | 13.60 |
| Feedback | Prescriptive | 47 | 100.00 | - | - | 66 | 100.00 |
| | Interrogative | - | - | 35 | 100.00 | - | - |

Note: *n*, sample; DI, direct instruction; TGA, tactical games approach; STBU, service teacher´s basketball unit; SSG, small-sided game.

Depending on the use that the in-service physical education teacher made of the pedagogical variables, it can be seen that in the STBU programme, there was a predominance of tasks without opposition, exercises as the teaching means, and prescriptive feedback to correct mistakes. Bearing in mind the scientific literature, these characteristics are closer to the traditional DI method [9].

### 2.5. Procedure

First, approval was requested for this study from the University Bioethics Committee (Ref. 247/2019). Authorisation was subsequently requested from the school management team and the physical education teacher to be able to conduct the study in the school.

Once the authorisation of the school management team had been obtained, informed consent was requested from the parents or legal guardians of the students. A meeting was organised with the participation of the researcher, in-service physical education teacher, management committee and students' families, where the objectives of the study and procedures to be followed were explained. Finally, they were given the informed consent forms which had to be signed and handed in for the students to take part in the study, following the ethical guidelines of the Declaration of Helsinki and Organic Law 15/1999 of 13th December on the protection of personal information (LOPD) (BOE 14th December 1999).

After the authorisations were received, an initial evaluation was conducted that consisted of a pre-test in which the students completed the TDPKB. This evaluation also included the following sociodemographic information: (1) school year; (2) age of the students, (3) years of basketball practice (1, 2, 3, 4 or more than 5); (4) how much they enjoyed playing basketball (1–10); (5) if they participated in any form of competition out of school (YES/NO); and (6) the level they considered they had as a player (1–10). Then, the DI, TGA and STBU intervention programmes were implemented randomly for nine sessions of one hour duration. Two more evaluation sessions were conducted using 3vs3 matches, increasing the intervention time to a total of 11 h. The volume of practice in the programmes was adequate, as interventions of more than eight hours are associated with the best results [19]. Finally, after the implementation of the programmes, a final evaluation was performed, a post-test, in which the students again completed the TDKPB.

The researcher explained the procedure to be followed to the students at the beginning of each test, so that it was clear how they had to complete them, and both tests had a maximum duration of one hour. During the implementation of the programmes, an audio was recorded of each session with the aim of reviewing the procedure and ensuring the adjustment of the intervention to the previously planned methods. In the case of the in-service teacher, the recording served to analyse the intervention teaching process.

### 2.6. Data Analysis

Firstly, criteria assumption tests were performed to identify the characteristics of the study data [47]. The Shapiro–Wilks and the Levene tests showed that the study variables complied with the assumption of normality and randomness, so that parametric mathematical models were used to test the hypothesis. A descriptive analysis was subsequently performed to determine the characteristics of the participants according to their previous experience in basketball. A factorial ANOVA for independent samples was carried out to compare the level of declarative and procedural knowledge among the different groups in the pre-test and post-test [47]. The differences among groups (methods) were identified with the *Bonferroni post-hoc test* for multiple comparisons.

A *t-test for related simples* was also performed to compare the level of declarative and procedural knowledge acquired by the students in each group after the implementation of the intervention programmes (post-test) with the initial values (pre-test) [47]. Subsequently, a factorial ANOVA and a *test of repeated measures* were performed to determine the effect of the variables, methodology and experience, on the students' declarative and procedural knowledge. The differences among groups according to the methodology, experience and

the interaction of these variables, were again identified with *Bonferroni's post-hoc* multiple comparison test.

Lastly, the effect size of the statistical analyses was determined using Cohen's d [48], partial eta squared ($\eta^2$) and observed power ($\phi$) [49]. For observed power, values (>0.80) were considered optimal. Cohen's d effect sizes were considered as follows: small (0.200–0.499), medium (0.500–0.799), and large (>0.800). Regarding ($\eta^2$), the range was small (0.010–0.059), medium (0.060–0.139), and large (>0.140) [49,50].

### 3. Results

The descriptive results for each intervention programme and each programme depending on previous experience in basketball are presented in Table 3.

**Table 3.** Descriptive results of the pre-test and post-test by programme and basketball experience.

| Method | Knowledge | Experience | n | Pre-Test $M \pm SD$ | Post-Test $M \pm SD$ | Post-PRE $\Delta$ |
|---|---|---|---|---|---|---|
| DI (6A) | Declarative | Yes | 8 | 12.38 ± 3.16 | 15.00 ± 3.07 | 2.62 |
| | | No | 9 | 9.33 ± 2.50 | 11.56 ± 3.47 | 2.23 |
| | | Total [1] | 17 | 10.67 ± 3.09 | 13.18 ± 3.64 | 2.51 |
| | Procedural | Yes | 8 | 8.38 ± 1.41 | 7.38 ± 2.33 | −1.00 |
| | | No | 9 | 7.56 ± 1.33 | 7.33 ± 2.06 | −0.23 |
| | | Total [1] | 17 | 7.89 ± 1.37 | 7.35 ± 2.12 | −0.54 |
| | Total | Yes | 8 | 20.75 ± 3.73 | 22.38 ± 4.57 | 1.63 |
| | | No | 9 | 16.89 ± 3.48 | 18.89 ± 4.86 | 2.00 |
| | | Total [1] | 17 | 18.56 ± 3.94 | 20.53 ± 4.91 | 1.97 |
| TGA (6B) | Declarative | Yes | 10 | 11.70 ± 2.54 | 13.30 ± 3.27 | 1.60 |
| | | No | 9 | 11.33 ± 2.78 | 15.11 ± 3.89 | 3.78 |
| | | Total [1] | 19 | 11.53 ± 2.59 | 14.16 ± 3.59 | 2.63 |
| | Procedural | Yes | 10 | 7.30 ± 1.68 | 7.40 ± 2.22 | 0.10 |
| | | No | 9 | 7.11 ± 1.62 | 7.89 ± 1.54 | 0.78 |
| | | Total [1] | 19 | 7.21 ± 1.65 | 7.63 ± 1.89 | 0.42 |
| | Total | Yes | 10 | 19.00 ± 3.97 | 20.70 ± 4.86 | 1.70 |
| | | No | 9 | 18.44 ± 3.18 | 23.00 ± 4.87 | 4.56 |
| | | Total [1] | 19 | 18.74 ± 3.53 | 21.79 ± 4.87 | 3.05 |
| STBU (6C) | Declarative | Yes | 5 | 12.40 ± 1.95 | 15.60 ± 4.88 | 3.20 |
| | | No | 13 | 11.46 ± 2.82 | 11.54 ± 3.21 | 0.08 |
| | | Total [1] | 18 | 11.72 ± 2.59 | 12.67 ± 4.04 | 0.95 |
| | Procedural | Yes | 5 | 7.20 ± 1.10 | 7.60 ± 1.14 | 0.40 |
| | | No | 13 | 6.08 ± 1.89 | 6.38 ± 2.63 | 0.30 |
| | | Total [1] | 18 | 6.39 ± 1.75 | 6.72 ± 2.35 | 0.33 |
| | Total | Yes | 5 | 19.60 ± 2.30 | 23.20 ± 5.45 | 3.60 |
| | | No | 13 | 17.54 ± 3.95 | 17.92 ± 5.19 | 0.38 |
| | | Total [1] | 18 | 18.11 ± 3.63 | 19.3 9 ± 5.65 | 1.28 |

Note: *M*, mean; *SD*, standard deviation; DI, direct instruction; TGA, tactical games approach; STBU, service teacher´s basketball unit; total [1], combined declarative and procedural without experience.

Table 4 shows the differences in the pre-test and post-test level of knowledge among the three groups. The results indicate that the students who participated in the DI programme presented higher initial levels of procedural knowledge (pre-test) ($p < 0.05$). In the multiple comparison analysis, the DI method group scored higher levels than the STBU group ($p < 0.05$). This shows that the starting situation of the groups was not the same regarding procedural knowledge.

**Table 4.** Declarative, procedural and total knowledge among the groups in the pre-test and post-test.

| Test | Knowledge | Method | $n$ | $M \pm SD$ | $F$ | $df$ | $p$ | $\eta^2$ | $\phi$ |
|------|-----------|--------|-----|------------|-----|------|-----|----------|--------|
| Pre-test | Declarative | DI | 18 | $10.67 \pm 3.09$ | 0.748 | 2 | 0.48 | 0.03 | 0.17 |
| | | TGA | 19 | $11.53 \pm 2.59$ | | | | | |
| | | STBU | 18 | $11.72 \pm 2.59$ | | | | | |
| | Procedural | DI | 18 | $7.89 \pm 1.37$ | 3.965 | 2 | 0.02 * | 0.13 | 0.69 |
| | | TGA | 19 | $7.21 \pm 1.65$ | | | | | |
| | | STBU | 18 | $6.39 \pm 1.75$ | | | | | |
| | Total | DI | 18 | $18.56 \pm 3.94$ | 0.139 | 2 | 0.87 | 0.01 | 0.07 |
| | | TGA | 19 | $18.74 \pm 3.53$ | | | | | |
| | | STBU | 18 | $18.11 \pm 3.63$ | | | | | |
| **Test** | **Knowledge** | **Method** | $n$ | $M \pm SD$ | $F$ | $df$ | $p$ | $\eta^2$ | $\phi$ |
| Post-test | Declarative | DI | 17 | $13.18 \pm 3.64$ | 0.477 | 2 | 0.48 | 0.03 | 0.17 |
| | | TGA | 19 | $14.16 \pm 3.59$ | | | | | |
| | | STBU | 18 | $12.67 \pm 4.04$ | | | | | |
| | Procedural | DI | 17 | $7.35 \pm 2.12$ | 0.882 | 2 | 0.42 | 0.03 | 0.19 |
| | | TGA | 19 | $7.63 \pm 1.89$ | | | | | |
| | | STBU | 18 | $6.72 \pm 2.35$ | | | | | |
| | Total | DI | 17 | $20.53 \pm 4.91$ | 1.004 | 2 | 0.37 | 0.04 | 0.21 |
| | | TGA | 19 | $21.79 \pm 4.87$ | | | | | |
| | | STBU | 18 | $19.39 \pm 5.65$ | | | | | |

Note: $n$, sample; $M$, mean; $SD$, standard deviation; $F$, factorial ANOVA; $df$, degrees of freedom; $\eta^2$, partial eta squared; $\phi$, observed power; DI, direct instruction; TGA, tactical games approach; STBU, service teacher´s basketball unit; * $p < 0.05$.

After the implementation of the programmes, no significant differences were observed among the groups ($p > 0.05$). However, the means indicate a tendency towards higher scores in the TGA group in the two types of knowledge.

Table 5 shows the level of declarative, procedural and total knowledge acquired by the students in each group after the implementation of the intervention programmes (pretest/post-test). The analysis of intragroup differences indicates that the groups that received the DI and TGA methods significantly improved their declarative and total knowledge ($p < 0.05$). Procedural knowledge did not improve significantly in any of the groups.

Figure 1 analyses the effect of experience and method on the different types of knowledge in basketball, both in the pre-test and post-test. The results show that the students from the different groups that practised basketball as an out of school activity presented higher levels of declarative, procedural and total knowledge in the pre-test and post-test. However, there is an exception, as the students from the TGA group that had not practised basketball previously, achieved higher levels of knowledge in the post-test than students who had practised.

A repeated measures analysis was performed to study the effect of the methodology and practice in the pre-test and post-test using the factors methodology and experience (Table 6). The results indicate that the quality of the declarative and total knowledge of the student was not the same at the two time points of the intervention, as learning had occurred. However, methodology combined with previous experience in basketball did not significantly influence learning.

**Table 5.** Level of declarative and procedural knowledge acquired in each group (pre-test/post-test).

| Method | Knowledge | *n* | *M* ± *SD* | *t* | *df* | *p* | *Cohen's d* |
|---|---|---|---|---|---|---|---|
| DI | Declarative pre-test | 17 | 10.76 ± 3.15 | −3.756 | 16 | 0.00 * | 0.71 |
| | Declarative post-test | 17 | 13.18 ± 3.64 | | | | |
| | Procedural pre-test | 17 | 7.94 ± 1.39 | 1.273 | 16 | 0.22 | −0.33 |
| | Procedural post-test | 17 | 7.35 ± 2.12 | | | | |
| | Total pre-test | 17 | 18.71 ± 4.01 | −2.214 | 16 | 0.04 * | 0.41 |
| | Total post-test | 17 | 20.53 ± 4.91 | | | | |
| TGA | Declarative pre-test | 19 | 11.53 ± 2.59 | −3.527 | 18 | 0.00 * | 0.84 |
| | Declarative post-test | 19 | 14.16 ± 3.59 | | | | |
| | Procedural pre-test | 19 | 7.21 ± 1.65 | −1.117 | 18 | 0.28 | 0.24 |
| | Procedural post-test | 19 | 7.63 ± 1.89 | | | | |
| | Total pre-test | 19 | 18.74 ± 3.53 | −3.508 | 18 | 0.00 * | 0.72 |
| | Total post-test | 19 | 21.79 ± 4.87 | | | | |
| STBU | Declarative pre-test | 18 | 11.72 ± 2.59 | −1.111 | 17 | 0.28 | 0.28 |
| | Declarative post-test | 18 | 12.67 ± 4.04 | | | | |
| | Procedural pre-test | 18 | 6.39 ± 1.75 | −0.825 | 17 | 0.42 | 0.16 |
| | Procedural post-test | 18 | 6.72 ± 2.35 | | | | |
| | Total pre-test | 18 | 18.11 ± 3.63 | −1.371 | 17 | 0.19 | 0.27 |
| | Total post-test | 18 | 19.39 ± 5.65 | | | | |

Note: *n*, sample; *M*, mean; *SD*, standard deviation; *t*, t-test for related simples; *df*, degrees of freedom; DI, direct instruction; TGA, tactical games approach; STBU, service teacher´s basketball unit; * *p* < 0.05.

**Table 6.** Intrasubject effects of the methodology and experience in the pre-test/post-test.

| Knowledge | Variables | *df* | *M* | *F* | *p* | $\eta^2$ | $\phi$ |
|---|---|---|---|---|---|---|---|
| Declarative | Time point | 1 | 125.87 | 25.981 | 0.00 * | 0.35 | 1.00 |
| | Time point * Methodology | 2 | 2.36 | 0.487 | 0.62 | 0.02 | 0.12 |
| | Time point * Experience | 1 | 1.25 | 0.259 | 0.61 | 0.00 | 0.08 |
| | Interaction | 2 | 14.45 | 2.983 | 0.06 | 0.11 | 0.55 |
| Procedural | Time point | 1 | 0.09 | 0.058 | 0.81 | 0.00 | 0.06 |
| | Time point * Methodology | 2 | 2.90 | 1.831 | 0.17 | 0.07 | 0.36 |
| | Time point * Experience | 1 | 1.28 | 0.811 | 0.37 | 0.02 | 0.14 |
| | Interaction | 2 | 0.44 | 0.275 | 0.76 | 0.01 | 0.09 |
| Total$^2$ | Time point | 1 | 132.74 | 20.148 | 0.00 * | 0.30 | 0.99 |
| | Time point * Methodology | 2 | 4.55 | 0.691 | 0.51 | 0.03 | 0.16 |
| | Time point * Experience | 1 | 0.00 | 0.000 | 1.00 | 0.00 | 0.05 |
| | Interaction | 2 | 18.89 | 2.868 | 0.07 | 0.12 | 0.54 |

Note: *M*, quadratic mean; *F*, repeated measures analysis; *df*, degrees of freedom; $\eta^2$, partial eta squared; $\phi$, observed power; interaction, time point * methodology * experience; total$^2$, combined declarative and procedural knowledge; * *p* < 0.05.

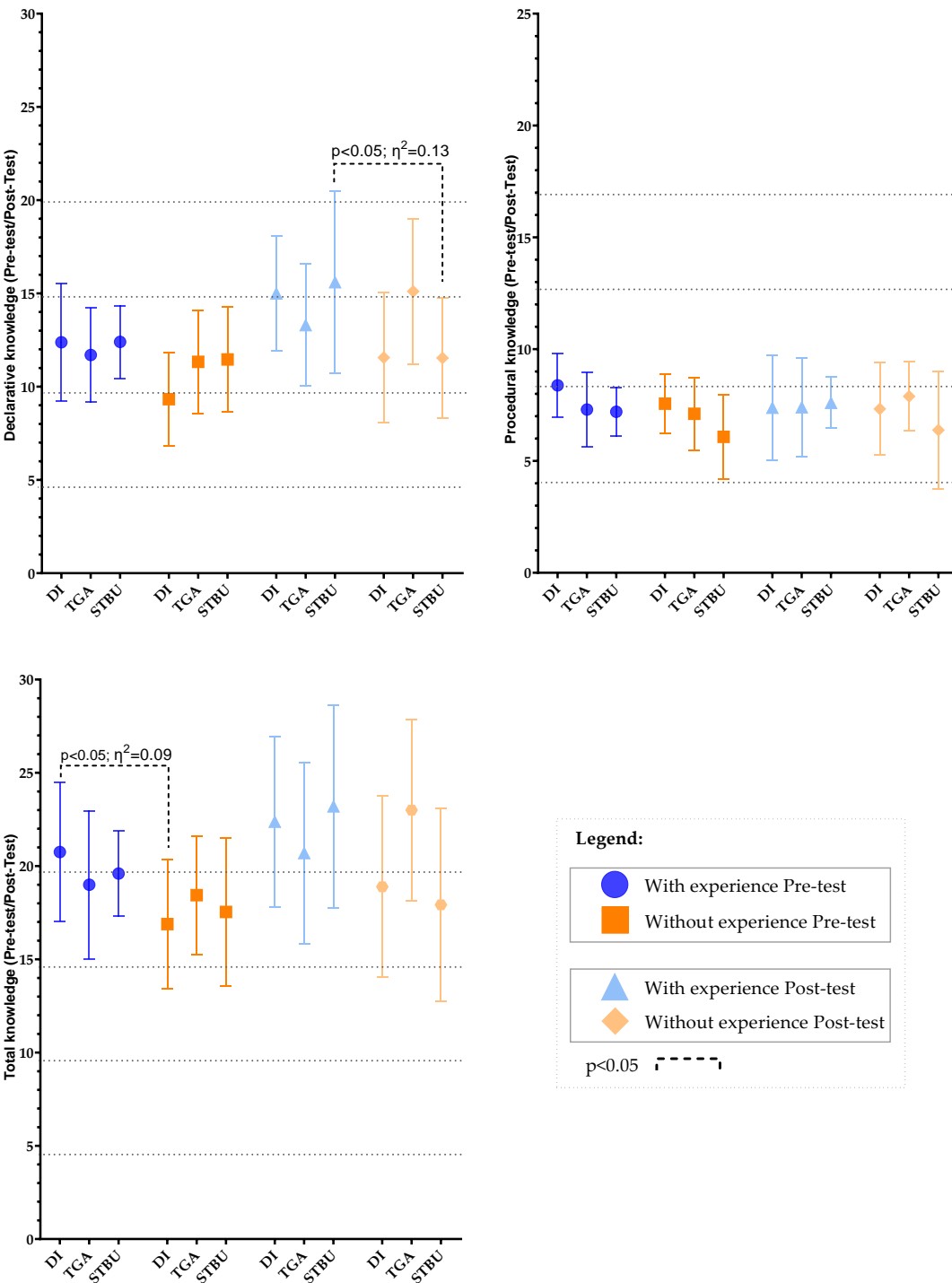

**Figure 1.** Diagrams showing the level of declarative, procedural and total knowledge according to the method and previous experience in the pre-test/post-test.

Table 7 analyses the effect of the methodology and experience according to inter-subject information. The results indicate that the level of declarative knowledge was not the same at the two time points of the intervention depending on the students' experience.



**Table 7.** Inter-subject effects of the methodology and experience in the pre-test/post-test.

| Knowledge | Variables | df | M | F | p | $\eta^2$ | $\phi$ |
|---|---|---|---|---|---|---|---|
| Declarative | Intersection | 1 | 15,682.63 | 1052.09 | 0.00 * | 0.96 | 1.000 |
| | Methodology | 2 | 6.37 | 0.428 | 0.65 | 0.02 | 0.11 |
| | Experience | 1 | 0.11 | 4.671 | 0.04 * | 0.09 | 0.56 |
| | Methodology * Experience | 2 | 39.85 | 2.674 | 0.08 | 0.10 | 0.51 |
| Procedural | Intersection | 1 | 5298.55 | 934.14 | 0.00 | 0.95 | 1.000 |
| | Methodology | 2 | 5.84 | 1.03 | 0.36 | 0.04 | 0.22 |
| | Experience | 1 | 5.80 | 1.02 | 0.32 | 0.02 | 0.17 |
| | Methodology * Experience | 2 | 3.57 | 0.63 | 0.54 | 0.03 | 0.15 |
| Total$^2$ | Intersection | 1 | 39,212.48 | 1265.436 | 0.00 | 0.96 | 1.00 |
| | Methodology | 2 | 4.95 | 0.160 | 0.85 | 0.01 | 0.07 |
| | Experience | 1 | 115.64 | 3.732 | 0.06 | 0.07 | 0.47 |
| | Methodology * Experience | 2 | 60.98 | 1.968 | 0.15 | 0.08 | 0.39 |

Note: *M*, quadratic mean; *F*, repeated measures analysis; *df*, degrees of freedom; $\eta^2$, partial eta squared; $\phi$, observed power; total$^2$, combined declarative and procedural knowledge; * $p < 0.05$.

## 4. Discussion

Traditionally, sports learning has been evaluated using closed tests or those of motor abilities [25]. According to the scientific literature, these type of tests have limitations for being applied to invasion sports, as they do not include decision making and real play during the development of the game [51]. Thus, the analysis of declarative and procedural knowledge of the sport has become the new tool to evaluate the cognitive sphere in invasion sports [33]. This investigation aimed to analyse the acquisition of declarative and procedural learning in primary education students according to the teaching–learning model implemented and their previous experience. After the administration of the different intervention programmes for basketball in the educational context, it was identified that the students evidenced improvements which were more significant in the students who received the PEAB programme, based on the TGA teaching method.

Players' knowledge of aspects such as technique, history or the rules of the sport has been evaluated using questionnaires or tests [37,52]. In these investigations, a non-linear pedagogical method with an SCA was used, influencing their understanding of the tactical aspects of the game and stimulating their reflection, and after the intervention it has been found that the experimental subjects improved their declarative knowledge. The majority of studies that use the SCA and also an instrument that evaluates declarative knowledge on the logic of the game, obtain significant improvement as a result [37,53]. Regarding procedural knowledge, conclusive results have not been found [19].

Specifically, the results of this study indicate that the programmes based on the TGA, DI and the STBU programme induced improvements in the levels of declarative and procedural knowledge in all the groups, except for procedural knowledge in the DI group. These differences were significant in declarative and total knowledge in the DI and TGA methods, but no differences were found in procedural knowledge. This could be due to the fact that the teaching they had received in previous years for learning invasion sports [54] had been more focused on technical aspects, without bearing in mind the strategies to be used during play and the difficulties for interpreting the most suitable tactics in each game situation, suppressing the students' creativity and decision making [29]. In the studies by García-Ceberino et al. [35] and Serra-Olivares et al. [41], the intervention programmes based on tactics also induced improvements in the levels of declarative and procedural knowledge. Sports teaching programmes based on a defined conceptual method result in a significant improvement in the students. The implementation of an undefined method, based on the selection of random tasks, such as the STBU, does not cause improvement in the students.

Similar studies carried out on invasion sports such as basketball [25], football [34] or handball [14] state that the teaching methods based on tactics induce higher levels

of knowledge. In the present study, after the implementation of the programmes, no significant differences were observed among the groups. However, the means indicate a tendency for a higher score in both types of knowledge after the TGA method. In different studies [36,55,56] differences were significant in favour of the tactical group regarding knowledge of the sport. However, in the study by García-Ceberino et al. [35], the students participating in the DI programme, based on technique, showed significant differences between the pre-test and post-test on declarative and procedural knowledge, while the students who participated in the TGA did not show significant differences. This was due to the heterogeneous distribution of the groups in physical education. Programmes of sports teaching in the school context based on TGA have been identified as the most suitable, as they result in the acquisition of learning related to the actual playing of the sport.

This study has analysed the effect of previous experience on the level of declarative and procedural knowledge acquired by students. Different authors [39,40] state that students possess different levels of knowledge according to their experience in sports practice and the main element that differentiates experts from novices is decision making [57].

In this study, 44.40% of the students that participated in the DI practised basketball out of school. In the TGA, 52.60% practised it, while in the STBU programme designed by the teacher, 27.80% of the students practised it. After the implementation of the programmes, the students from the TGA group who had not previously practised basketball achieved higher levels of learning in declarative and procedural knowledge compared to the students who did practise it. Furthermore, these students also achieved more learning than the students without experience in the rest of the programmes. Therefore, the students who had not practised basketball previously were able to better understand the sport through a teaching method based on the comprehension of the sport. Different studies identify improvements in cognitive type variables related with decision making and tactics using this methodology [20,36,58]. García-Ceberino et al. [35] identified that the students who did not have previous experience improved regardless of the method, and those who did have experience improved even more with the TGA programme. González-Espinosa et al. [24] analysed the differences in basketball learning depending on the methodology and gender of the students. After the intervention, the girls and boys from the TGA programme recorded better results than those obtained with the DI method. Moreover, the girls achieved more learning than the boys with the TGA.

The method and experience of the students affected some levels of declarative and procedural knowledge in the pre-test and post-test. Specifically, experience affected the initial knowledge of the sport, as the students who had practised basketball previously obtained better scores than the students who had no experience, which is logical. The same results were found in similar studies. In the studies conducted by García-Cebrino et al. [35] and Serra-Olivares et al. [41], the more experienced students recorded better results. Thus, practising sport outside the educational context positively influences the student's initial level of declarative and procedural knowledge. It has been demonstrated that the improvement of inexperienced students is significantly greater, showing that the teaching programmes are valid, permitting students without knowledge of a sports content to improve.

The interaction between the factors of methodology and experience is significant among the groups regarding declarative knowledge in the post-test. This suggests that the effect of practice interacts with some of the levels of the method; specifically, with the DI and STBU methods, as the students obtained very similar scores. However, the effect of the methodology combined with the previous experience in basketball did not significantly influence learning. Previous experience has been more relevant for declarative knowledge than for procedural knowledge. Regardless of basketball experience, students have more facility to transfer procedural knowledge obtained from other invasion sports. Therefore, no significant differences are obtained between pre-test/post-test. In procedural knowledge, students with experience of the DI method obtain less learning in the post-test;

however, students without experience obtain more knowledge in the post-test through the TGA.

With respect to the methodology, in the descriptive analysis of the different intervention programmes it can be seen that the DI and STBU programmes make a similar use of the different pedagogical variables. Both programmes focus on working on tasks without opposition using exercises, and the teacher uses prescriptive feedback to communicate with the students, characteristics of a traditional methodology [13,59]. However, the intervention programme based on the TGA focuses on the use of tasks with opposition, especially in individual play (1vs1) and in small-sided games with numerical inequality. It does not emphasise execution but understanding the sport using specific games and interrogative feedback [60,61]. The characteristics of the sports discipline to be taught should condition the teaching method. Invasion sports, open sports involving a lot of decision making and the presence of opponents, should be taught with SCA methodologies.

According to Kirk et al. [62], to acquire the principles of play, a process of understanding needs to be produced thanks to the active participation of the learners. For this, specific and contextualised tasks are used from the sport itself [14] involving opponents and presenting a challenge for the students. The use of interrogative communication on the part of the teacher/coach provokes reflection in the students which will allows them to construct more complex semantic networks in their brains, related with the information they already possessed. The teachers/coaches therefore use questions that guide the players, allowing them to give meaning, significance and functionality to the "how" of their actions [10,63]. Therefore, the tasks in the TGA have given the students a more active role in the teaching–learning process, achieving greater declarative and procedural learning in the sport of basketball.

## 5. Conclusions

The students who train using tactical methods, TGA, at the end of the teaching–learning process, present greater knowledge of the sport than participants who learn according to the DI and STBU methods.

The students who follow defined teaching methods, DI and TGA, improve after their implementation, while those that follow the STBU method do not. The magnitude of the improvements of the students who are administered a TGA-based programme is greater than that of those who follow the DI method. Therefore, when there is a conceptual foundation in the design and planning of the tasks, better learning is achieved than when there is no conceptual foundation, e.g., STBU. Moreover, students who learn with the TGA method acquire more knowledge than those who follow the DI method.

Experience is a determinant factor in the teaching–learning process. The students with previous experience of basketball have more initial knowledge of the sport. These differences are significant in initial and final knowledge when experience interacts with the teaching method based on direct instruction, as once the intervention programme finishes, the students without experience present limited evolution in their learning using these types of methods. Therefore, planned sports teaching close to the direct instruction method (DI and STBU) increases the differences in the learning of the students according to previous experience. The students without sports experience improve more when a training programme based on TGA is administered.

The results indicate that the use of an intervention programme based on tactics, the TGA, that seeks an understanding of the logic of the game and stimulates reflection in the students, is effective for improving their declarative and procedural knowledge. Moreover, the TGA is identified as a method with which the students without experience learn more and more quickly, as they are able to better understand the sport through a method based on understanding the game.

*5.1. Practical Application*

This study provides important information for physical education teachers, as it compares the effects produced by different teaching methods and previous experience on students' learning, permitting them to discover which methodology most favours the teaching–learning process in invasion sports. The data obtained are important both for the educational field and for sports initiation training and invite them to take the previous level of the students into consideration in the planning of the learning process.

*5.2. Limitations*

Among the limitations to this study, it should be highlighted that the results obtained came from a sample with concrete characteristics, and the duration was not sufficiently long to determine large significant differences. Thus, further research is necessary to extend the duration of the intervention and include more participants to vary the profile of the subjects with the aim of providing more information and achieving greater experimental control.

**Author Contributions:** Conceptualisation, M.G.G., S.J.I., and S.F. Methodology, M.G.G., S.J.I., and S.F. Formal analysis, M.G.G. Reviewers, J.M.G.-C., S.J.I., and S.F. Writing—original draft preparation, M.G.G. Writing—review and editing, J.M.G.-C., S.J.I., and S.F. Visualisation, M.G.G. Supervision, J.M.G.-C., S.J.I., and S.F. All authors have read and agreed to the published version of the manuscript.

**Funding:** This study has been partially subsidised by the Aid for Research Groups (GR18170) from the Regional Government of Extremadura (Department of Employment, Companies and Innovation), with a contribution from the European Union from the European Funds for Regional Development.

**Institutional Review Board Statement:** The study were conducted according to the guidelines of the Declaration of Helsinki and Organic Law 15/1999 of 13 December on the protection of personal data (BOE, 298, 14 December 1999) in order to guarantee the ethical considerations of scientific research with human subjects. The approval of this study was requested from the Bioethics and Biosafety Committee of the University of Extremadura. It was positively valued with reference number (Ref. 247/2019) on 18 December 2019.

**Informed Consent Statement:** Informed consent was obtained from all subjects involved in the study.

**Data Availability Statement:** Not applicable.

**Acknowledgments:** The authors would like to thank the physical education teacher for participating in this investigation.

**Conflicts of Interest:** The authors declare no conflict of interest.

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
