# Peer review of "Analysis of Declarative and Procedural Knowledge According to Teaching Method and Experience in School Basketball"

_sustainability, doi:10.3390/su13116012_

Round 1
Reviewer 1 Report
Analysis of declarative and procedural knowledge according to 2 teaching method and experience in school basketball
The subject of this research work is very interesting. The authors aim is to compare the acquisition of declarative and procedural knowledge after the implementation of several intervention programmes in school basketball, according to the methodology and prior experience of the students
Paper strengths:
The research is ambitious and has important practical implications. The procedure is correct and statistical analysis rigorous. The bibliographical references are current
Paper weaknesses:
There are many results that are not discussed in depth. I think the authors should try to explain better why experience is more relevant to declarative rather than procedural knowledge, and how it interacts with educational programmes. There is also little comment on the similarities between the DI programme and the STBU and the results obtained.
I believe, therefore, that the discussion should be developed a little further.
Reviewer 2 Report
Thank you for a chance of reviewing this article. Although it is intersting study I have some comments and suggestions.
Abstract is concise and informative.
Regarding the Introduction
in the paragraph where the authors are describing the models of teaching invasions games they provide two models with suggestion that TCA is linked with Direct Instuction, while SCA is linked with tactical games approach - in my opinion there is a mixture of concepts - TCA is linked with DI (and I can understand that implications), however, I am not sure why you linked SCA with TGA only? First of all - opposite to DI is Indirect Instuction approach with methods and teaching styles like guide-discovery and problem-based learning - Why haven't you refer to that? Also with SCA there are other methodologies that could be employed, specifically in games - like for example Co-operative learnig or Health(a)warenees modular model learning. You could also look at Bunker and Thorpe's Teaching Games for Understanding approach (look into : Navigating the Benefits and Challenge of the Teaching Games for Understanding Model).
All of these mentioned methodologies are linked with non-linear pedagogy, which should also be mentioned in your Introduction (look at : Nonlinear pedagogy: a constaints-led framework for understanding emergence of game play and movement skills")
So you should strenghten your rationale why you made your selection and associations and links between teaching/learning concepts.
Also in the Introduction there is little on the specificy of invasion games (for example its differences in the intensity loads in various phases of the teaching/learning process - look for example into : The effects of two instructional formats on the heart rate intensity and skill development of physical education students and to indicate that the learning phase also matters on the effectiveness and perception of both sport skills as well as sport-related knowledge look into "Health-related intensity profiles of Physical Education classes at different phases of the teaching/learning process" and ...Acquistion of sport knowledge and skill: the role of self-regulatory processes") to provide a broader picture and rationale for undertaking study like yours. All these references can also be used in Discussion.
In the methods section
the procedure is described in a clear manner. So are also the research tools with reliability co-efficents provided which is essential and that is good.
In Results - I have to admit that although tables and figure are neat and well-described tha amount of data makes this part difficult to follow.
There is a lot of statistical data and I have an impression that it does not help in conveying the message from the study - Usually in the papers authors should avoid doubling the same results in tables and figures - maybe you should consider leaving out some date which is not entirely essential for the purpose of this article.
In Discussion in line 355-356 the authors report on the improvements in the group of students who develped TGA model - and I have to admit this is confusing. How were the students resonsible for the development of the TGA model ? Isn't it the role of the teacher and the pupils follow the contenst which has been prepared by the teacher accordingly to the theoretical frameowork of the model?
In this section the authors refer also to some other studies (some by the authors themselved) regarding different sports - why to study over and over the same procedure on similar fields (invasions games)? You need to provide arguments why you wanted to continue this kind of research line - what was missing in the previous studies ? And this should have been included into Introduction section.
Generally, Discussion seems too long and there are too many problem-areas tackled - it is like too mamy mashrooms in a soup. Maybe you should leave the links with professional sport or neurological activation for another occasion.
References
the list of references is impressive, but very much Spanish-authors based (with a lot of self-citations) and this would have to be balanced.
Round 2
Reviewer 2 Report
I am glad you have included comments and suggestions, but please check the accuracy of referencing concerning positions 8 or 9 on non-linear pedagogy both in text and in the references.
